# Design Science and Co-Designing of Hybrid Workplaces

**Marko Lahti** [1,*] and **Suvi Nenonen** [2]

1. Department of Computer Science, Faculty of Technology, University of Turku, 20500 Turku, Finland
2. Department of Civil Engineering, Faculty of Built Environment, Tampere University, 33720 Tampere, Finland; suvi.nenonen@tuni.fi
* Correspondence: mklahti@utu.fi

**Abstract:** Background: Future places for learning and working are digitally and physically integrated hybrid environments. The archetypical context of learning is the classroom, and context of working is the office; especially in knowledge work. New information and communication technologies enable the spatial reconfiguration of work opening possibilities for work to take place across multiple locations. This paper aims to explore how the conceptual framework of design-science research in Information Systems can be applied when the design object is a hybrid working environment. Methods: The case study method as a qualitative approach was chosen; because it involves an empirical investigation of a particular contemporary phenomenon within its real life context using multiple sources of evidence. The empirical analysis of two hybrid working environments is based on Action Design Research (ADR)-entry points; where one analyzes two case studies stage by stage. By analyzing various stages in both case studies; one can identify co-designing challenges of hybrid working environments. Results: The results present four recommendations for co-designing of hybrid working environments. The use of hybrid working environment; the design of spatial solution; the identification of iterative processes; and the user experiences of presence and distance are significant. The Entry Point Analysis-tool can be used and further developed in analyzing and developing hybrid working environments. Conclusion: The results contribute to the tradition of usability studies. The usability briefing approach can be further developed by identifying the iterative processes inside the linear project management models. Additionally, design science research can find new insights from identification of the large stakeholder iterations more precisely.

**Keywords:** action design research; entry point analysis; project management; usability briefing; hybrid working environment; co-design; co-working

## 1. Introduction

Future places to learn and work are a digitally and physically integrated hybrid environment. The aim of this paper is to investigate the co-design processes of physical and digital solutions. Usability of built environment relies on the different service design methods. More of them can be found from the Information Systems field. Digital solutions are developed by applying the Action Design Research (ADR) process model by Mullarkey and Hevner [1]. The aim is to explore how the conceptual framework of design-science research in Information Systems can be applied when the design object is a hybrid working environment.

Instead of existing work and learning environments there is a shift towards more hybrid work and learning environments. This has been influenced by a number of different benefits [2,3]. Constraints have been the technology available and the courage for radical reform. Hybrid learning environments make it possible to combine physical, digital, and social learning in a novel way [4]. In this study, a process model for co-designing the hybrid work environment is presented to combine a physical space and digitalization in the design phase. More specifically, to solve the iterative process challenge for co-designing with users and co-designing with digital and physical design stakeholders. For the design

of physical environments, Fronczek-Munter [5] has introduced a usability briefing model that describes the development cycles of space, whereas Mullarkey and Hevner [1] have introduced an ADR process model based on the Information System field. The former model is based on the long-term conceptual research of usability of workplaces, while the latter focuses on usability of digital solutions. The Entry Point Analysis-tool is developed for the framework.

The paper consists of four main sections. The introduction presents the background of the study, the goal of the research, and then usability approaches. The hybrid environments for learning and working are discussed. The co-creation processes of usability briefing and action design research are presented and compared. Section 2 continues with methods, the research process, and case study descriptions. Additionally, the Entry Point Analysis-tool is presented. In Section 3, we present analysis of two case studies and results. Finally, Section 4 concludes the research by evaluating the limitations and proposing future research topics.

### 1.1. Hybrid Learning and Working Environments

The archetypical context of learning is the classroom, and context of working is the office, especially in knowledge work. Novel technologies enable the spatial reconfiguration of work giving possibilities to work in multiple locations [6]. One way to implement this is to have a live 3D constructed feed from a chosen space, where remote users can participate with local users in a virtual environment. Changes in educational practice are driving the emergence of hybrid learning environments [3]. Established educational methods change, expand, and replace established roles, resources, and locations. Moreover, the working environments need to follow the learning environments and provide more integrated solutions for efficient use of both digital and physical working environments.

What do we mean by hybrid in the context of learning and working environment? Tynjälä, et al. [7] described modes of learning, which can generally divide between learning that is situated in a working environment and an educational environment. The first one is mostly informal learning, whereas the last one is more formal. Moreover, Tynjälä et al. [7] identified a hybrid form of learning when learners worked collaboratively while using project-based learning. Herrington and Herrington [8] introduced authentic learning, which is a similar concept to project-based learning, where the idea is how knowledge is used in real life and providing activities reminiscent of activities in practice. In addition to this, Van Merriënboer, et al. [9] have noted a similar concept called authentic learning, where real life tasks are the driving force of the learning. One of the trends was to integrate Information and Communication Technology (ICT) in real life tasks or projects according to van Weert and Pilot [10], which has continued to this day. Goodyear et al. [11] pointed out how ICT can also promote a socio-cultural aspect, since it enables learners and teachers to collaborate and learn together from a distance. Zitter et al. [12] introduced a descriptive model of learning environment. Moreover, Zitter et al. [13] positioned the learning tasks mentioned by Van Merriënboer et al. [14] to the previous model. Zitter et al. [13] also points out that learning environment is formed by these concrete learning tasks. To describe the learning task, four different perspectives are distinguished, which are:

1. Agency perspective, to describe the roles of the participants;
2. spatial perspective, to study learning tasks on the physical and digital space;
3. temporal perspective, describing the needed time for the learning tasks;
4. instrumental perspective, important boundary objects to deliver intermediary and final results of the learning tasks.

The Organization for Economic Co-operation and Development (OECD) [15] presented Education Working Papers in 2012, where they applied the same model for different cases like technology, hospitality, and sports.

Chen and Chiou [16] studied results which indicated that students of hybrid learning environments felt a stronger sense of community than students in traditional environments. Besides that, the students had significantly higher learning results. Sonntag et al. [4] study



highlighted that augmented reality provides an opportunity to integrate physical, digital, and social learning in hybrid learning environments, thus enhancing learning interaction, motivation, and collaboration. Ibáñez et al.'s [17] study was aimed at system architecture and usability of a proof-of-concept for hybrid learning environments. It pointed out that usability had positive engagement effects on participants while participating in a 3D virtual mirror of the real space.

Halford [18] argued in his study that spatial hybridity changes the nature of the work, organization, and management across domestic, organizational, and digital space. The paper explored the implications of hybrid workspace. Halford [18] studied a financial service company that allowed part-time homeworking. The results were positive and changed, for instance, how people work, and finally Halford [18] concluded that previous studies have indicated that full-time home-working causes negative experiences and consequences, but material gathered in her study suggest that a combination of work spaces gave positive feedback. Later, co-working was defined as creative cities or districts, where two inter-linked tendencies are embedded together [19]. Marchegiani and Arcese [20] addressed collaborative spaces and co-working as hybrid workspace in their work and concluded it to be effective in the context of a collaborative and sharing economy. The conclusion was based on Gandini [21] and Bostman and Rogers's [22] earlier work. Marchegiani and Arcese [20] continued to say that the co-working space seems to give workers an area that supports the physical and the digital interaction simultaneously, which eventually leads to innovative outcome.

While the concept of hybrid learning and working environment can be defined in different ways, Hilli et al. [23] presented five designing, developing, and implementing principles for it in a higher education context. These principles are one way to divide and structure the design processes and practices while designing the hybrid learning space. All the five principles are further discussed by Hilli et al. [23]. Both Stommel [24] and Hilli et al. [23] present a hybrid pedagogy as a methodological approach for interlinked practices and processes. The challenge in both working and learning environments is to ensure that their usability is considered in the interaction of people, building, and technology.

### 1.2. The Goal of the Research

The aim is to explore how the conceptual framework of action design research (ADR) in Information Systems can be applied when the design object is a hybrid working environment. The research question asked is: How can the action design research paradigm be applied in understanding the usability challenges in hybrid working environments? The case study method was chosen as a qualitative approach, since it involves empirical research on a particular contemporary phenomenon in its real life using multiple sources of evidence [25]. The data used in the study is collected through interviews, participant workshops, and retrospective analysis of documents. This paper will enlarge the discussion in terms of hybrid working environments integrating people, organizations, technology, and buildings.

### 1.3. Usability Approaches in the Context of Physical and Digital Environments

#### 1.3.1. Usability Briefing Approach

Based on a series of studies on usability in the built environment conducted in Europe, which propose that management or governance of use-centric processes is seen as crucial to ensure not only functional, but also usable, outcome of co-designed built environments [26].

Incorporating the users' knowledge and preferences in the architectural, engineering, and construction (AEC) project is important [27]. Based on these studies, several process descriptions have been developed further. They emphasize the significance of user participation in different phases of the process, indicating many simultaneous processes.

A usability briefing model provides an overview of the activities in the usability briefing process. Its meaning as a continuous and dynamic process of capturing user

perspectives throughout all the phases of building projects is captured. The model is generic and simple to use, and it is meant to be used in the planning of new complex building projects [5]. Figure 1 illustrates how Fronczek-Munter [5] has introduced the usability briefing.

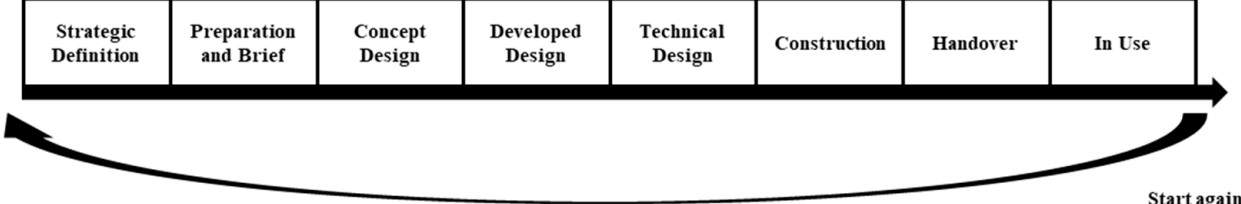

**Figure 1.** Usability briefing process modified from Fronczek-Munter [5].

The usability briefing's first two phases are preliminary to the project. At these stages, decision-makers create a shared vision. It takes into account, e.g., strategy objectives, data collection, organization, and order of priority of decisions. The next four phases are about design and construction. More specifically, it includes, e.g., an architectural vision, usability, innovation, usability, co-learning, co-designing, layout, functionality of design proposals, and maintaining usability. The last two phases are about moving in successfully, learning to use the facility, evaluations, requirement tests, and improvements [5]. During these phases, the user involvement, design, evaluations, and briefing activities ratio varies. These activities should interact with each other.

While the usability briefing model unites the different activities, it is a straightforward process. It is recommended that topics of all activities are well formalized, which make them easy to discuss in the meetings. When using the model, the focus should always be on usability in every phase. After the final phase, one can start the process again based on the evaluation and user experience.

### 1.3.2. Action Design Research Approach

Usability research in man-machine interaction has a long tradition. In comparison to the usability of a built environment, the usability of a digital environment has differences in scales of the object. Traditionally, in the development of information systems, two paradigms characterize much of the research in the field of information systems research: Behavioral science and design science [28]. To predict or explain human or organizational behavior, one can seek to develop and verify theories by using a behavioral-science paradigm. The paradigm of action design research tries to expand the boundaries of human and organizational features by creating new and innovative objects. Similarities can be found from the usability of the workplaces studies: One needs to understand the human-building relationship [29]. Hevner et al. [28] states that two paradigms, the behavioral and design science, are based on Information System (IS) science, which is at the intersection of people, organizations, and technology. Co-designing hybrid environments has multiple stages, which differ from each other (see Figure 2).

The entry points of each stage are described as follows [1]:

1. Problem-centered, to describe the research problem and specify where the solution is aimed for.
2. Objective-centered, to study design possibilities and in that way increase the data for the solution area.
3. Development-centered, aims to implement a designed solution that solves the research problem.
4. Observation-center, studies implemented a design and evaluates its usage to enhance the implementation even further.

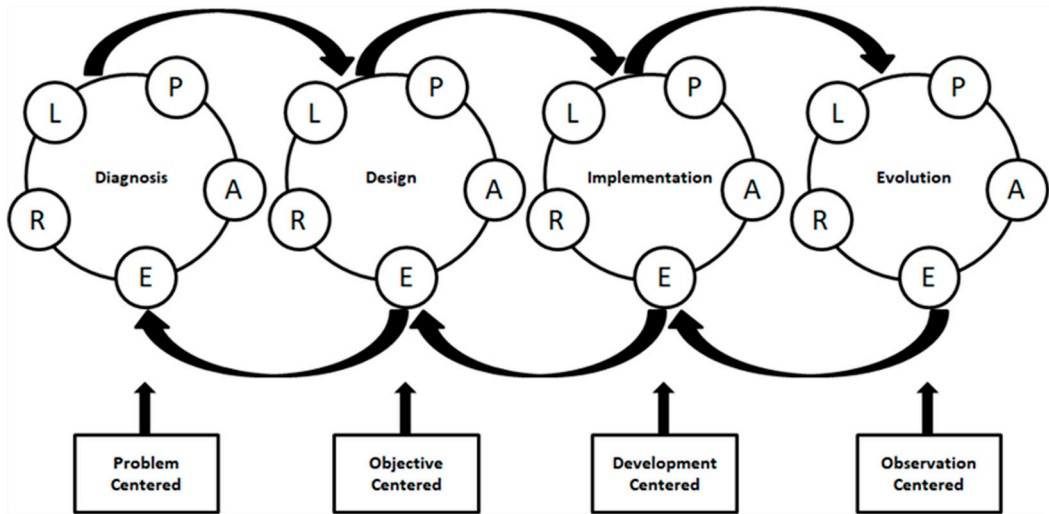

**Figure 2.** Action design research-process model with research entry points [1].

Mullarkey and Hevner [1] elaborated on ADR-entry points, which describe the starting point to use the ADR-study. Mullarkey and Hevner [1] even argued that the group of researchers had an obligation to recognize and present the entry point that motivates the ADR-study. The research entry point could occur at any one of the ADR-stages, as shown in Figure 2.

### 1.3.3. Similarities and Differences of Usability Briefing and Design Science Approach

To summarize, both models have similarities and differences (see Table 1).

**Table 1.** Comparison of the process models.

| Features | Usability Briefing | Action Design Research |
|---|---|---|
| Process | Aligned with the construction process | Aligned with the design process |
| User involvement | Strongly emphasized | Strongly emphasized |
| Identified phases | Eight phases | Four phases with five sub-phases |
| Iteration Frequency | Feedback loop from use of the building to begin of the new usability brief | Each phase includes iterations |

Both usability briefing and action design research approaches focus on the process. While usability briefing is aligned with the building lifecycle with different stake holders in the AEC-project, action design research focuses on process, which is an interaction with the external environment and research knowledge. User centricity can be seen at different stages in both models. Users are influencing to the matching of the physical or digital solution to the needs of the users.

The major differences between the approaches are in the length and rhythm of the process. The cycle of usability briefing is longer. There are no feedback loops within different phases, while in the ADR-model, it is visualized that iterations are taking place more frequently. This makes the ADR-model look more agile by the nature: Iteration steps are small and fast. The potential learning occurring among co-design stakeholders is visualized differently in models.

The co-design of hybrid learning and working environments needs to integrate the agile digital design sprints and more static progress of the built environment. In response to this need, this research tests the Entry Point Analysis-tool in two case studies of hybrid working environments.

## 2. Methods and Research Design

### 2.1. Case Study Approach

This research follows exploratory study principles as it aims to gain more understanding on design processes of hybrid working environments. The case study method as a qualitative approach was chosen, because it involves an empirical investigation of a particular contemporary phenomenon within its real life context using multiple sources of evidence [25]. Case selection criteria were the following:

1.  Work environments of about the same size;
2.  work environment in an academic work context;
3.  user participation in the co-design process;
4.  hybrid work environment in a significant role.

Two cases were selected: Case 1 was Health Tech Hub at the University of Turku in Finland. The size of the hybrid work environment was 187 sqm space. The research group of health technology relocated from the faculty building to the building of medical faculty of the university. Additionally, the building was used by researchers from the local hospital district and the university of applied sciences. The vision was to conduct health technology research in the research surroundings of medical sciences. Additionally, the research group had a lack of office space in the existing faculty building.

The second case was from the same university. The Future Tech Lab is 200 sqm space for open innovation in software engineering education and research. It was set up by the Finnish University of Turku to the main campus of the University of Namibia. The expected users were the students of future software engineering study programs, researchers, and supervisors. The vision was to develop a platform for the new innovations, which have global impact, and support research, development, and learning in open innovation processes.

Both cases represented the need to co-design both technical and physical solutions in an integrated manner to achieve the hybrid solutions. Table 2 represents the cases at a glance.

**Table 2.** Summary of the cases.

| Features | Case 1: Health Tech Hub | Case 2: Future Tech Lab |
| --- | --- | --- |
| Size of the space | 187 sqm | 200 sqm |
| Previous use of the space | Business premises | Storage and IT class room |
| New use of the space | Research and collaboration | Co-work and co-learn |
| Hybrid factors | Collaborative use of technology in ABW environments | Novel technology for overseas collaboration in flexible space |
| The initiator of the process | Research group | Management of two universities |
| Location | Turku, Finland | Windhoek, Namibia |
| User groups | Research group and partners | Research group and industry partners, Finnish and Namibian students and staff of universities |
| Estimated number of users | 30–35 | 40–50 |

The data gathering focused on two topics: The co-design process of the hybrid working environment and the user experience of the hybrid working environment. The aim of the study was to discover how to integrate the design of digital and physical to achieve a hybrid working environment that is usable. The qualitative approach to the problem seems most relevant because the co-design of either physical or digital environment is more typical than the integrated approach. The quantitative data is not easy to gather because the hybrid working environments are not yet the usual case in the context of working.

The data collection was performed in three steps.

The first step was based on literature review. It was a basis to the integrated Entry Point Analysis (EPA)-tool. The tool aims to identify the different phases of co-designing the hybrid working environments.

The second step was dedicated to the analysis of the co-design processes. The data was gathered by participatory workshops and document analysis. Four participatory workshops (see Table 3) were conducted in case 1 during 2018–2019. In the first case study, the participants were the research group members as users, interior designer, digital designer, and facilitator with co-design expertise. The topics of the workshops were:

1. Vision;
2. user profiles and needs;
3. iteration of the digital and physical solution;
4. iteration 2 of the digital and physical solution.

**Table 3.** Participatory workshops.

| Features | Case 1: Health Tech Hub | Case 2: Future Tech Lab |
|---|---|---|
| Amount of Participatory workshops | 4 | 5 |
| Participants | Research group members as users, interior designer, digital designer and facilitator with co-design expertise | Management from both universities, architect, two digital designers, employees from both universities and facilitator with co-design expertise. |
| Participants in average | 25 | 7 |
| Time schedule | 2018–2019 | 2019–2020 |
| Data | Process descriptions, Layout plans, minutes and other notes | Process descriptions, Layout plans, minutes and other notes |

In case two, five participatory workshops followed similar themes during 2019–2020. Participants of the workshops included the management from both universities, an architect, two digital designers, employees from both universities, and a facilitator with co-design expertise.

In the third step of data gathering focused on user experiences, data was gathered by observations and experience mapping questionnaires. Observations were conducted in case 2 two times after the retrofitted environment was in use. The researcher used an observation sheet to gather data from the usability of the solutions. User-questionnaires were conducted in both cases (Table 4).

**Table 4.** Experience mapping questionnaires.

| Features | Case 1: Health Tech Hub | Case 2: Future Tech Lab |
|---|---|---|
| Conducted questionnaires | 1 | 1 |
| Participants | Users of the hybrid working environment | Event organizers of the hybrid working environment |
| Responses | 8 | 6 |
| Amount of questions | 26 | 32 |
| Structure of questionnaires | Semi-structured questionnaire with open comments | Semi-structured questionnaire |
| Topics | User experience, development proposals and background information | User experience, development proposals and background information |

In the first case study, user experiences were gathered from users of the hybrid working environment by a feedback questionnaire with 26 questions, and the responses were gained from eight participants. The feedback questionnaire focused on user experiences in previous and current workspace as well as on needs for improvement. The more detailed topics concerning hybrid working environments were the availability of facilities, the need for change, teamwork, technical implementation, and opportunities for cooperation between stakeholders.

In the second case study, a feedback questionnaire with 32 questions provided insights from six event organizers that used the hybrid working environment. The focus was more about usability of the hybrid working environment for different purposes. The topics

were: How different spaces support collaboration and individual work and the factors that improve these, as well as how well hybrid environments have been implemented and what still needs to be taken account. Both Likert-scale questionnaires had similar statements, but the amount of open questions differed.

The intention of the questionnaires was not to produce generalized quantitative data, but serve as one sources of data next to the observations, participatory workshops, and document analysis. Triangulation [30] was used to complement the data and to find new information, in other words, to get additional pieces to the overall picture of hybrid working environments.

Because the response rate was small, some interviews were also conducted about the usability of the facilities for different purposes, benefits of the hybrid working environment, and the development ideas for the technical setup. The observations also provided data about the usability of the places.

### 2.2. Data Analysis

The data was organized in transcripts. To analyze the process more thoroughly, the Entry Point Analysis-tool based on Action Design Research approach was developed. The tool is applied from Hevner's theoretical model. The development was made in three co-design workshops among researchers who use the in digital context and researchers who are familiar with usability of the built environment.

Entry Point Analysis (EPA)-tool aims to identify the different phases of co-designing the hybrid working environments. It clarifies how to integrate the digital and physical environment to support user. It is also a tool to model the co-design hybrid learning environment. It is needed to improve the design processes of the digital and physical in a systematic way. The focus is on co-designing the artifact, which is a hybrid working environment. The EPA-tool maps four co-design stages with a five-step process in each stage (see Table 5) In this study, the data of two case studies was used in analysis.

**Table 5.** The Entry Point Analysis (EPA)-tool for co-design in the hybrid working environments.

| Stage | Hybrid Vision | Hybrid Integration | Hybrid Fit | Hybrid Fix |
|---|---|---|---|---|
| Main questions | What is the vision of the hybrid solution? <br><br> Who are involved? | What are the functions supported by digital platform—what are the requirements of this to physical environments? <br><br> Who are involved? | How does the digital and physical solution fit together to ensure usability? <br><br> Who are involved? | How is the hybrid working environment used and continuously both evaluated and developed? <br><br> Who are involved? |

In the hybrid vision stage, the hybrid working environment is seen as an integrated artifact which is supporting the collaboration and individual work tasks of the user. The main topic to co-design is the vision: For what purpose the hybrid working environment is developed. The artifact is developed in collaboration with different user groups and project stakeholders, and it leans on a diverse knowledge base. The questions asked in this stage are:

1. Why does one need to make or modify a hybrid environment?
2. What problems will the hybrid space solve—which functions it will support?

In the hybrid integration phase, the artifact is co-designed by finding solutions to the visions and functions which are identified in the first stage. The main topics on co-design are:

1. The requirements of individual and collaborative work for the digital environment;
2. the requirements of functions and digital environment to physical environment.

The knowledge base is formed by dialogue among users, digital experts, and built environment experts. The questions asked in this stage are:

1.  Which individual and collaborative functions need to be supported?
2.  Which technology supports them?
3.  What requirements are set for the physical environment based on functions and technology?

In the hybrid fit stage, the alignment of digital and physical environments is conducted, and the emphasis of co-design is on avoiding usability misfit. To co-design hybrid fit, one needs to ask:

1.  How is the hybrid working environment implemented?
2.  How is the solution evaluated?

In the hybrid fix stage, users can give feedback to develop the solution even further or if, for instance, the user's practice changes significantly, it might affect the current solution quite a bit. The questions asked in this stage are:

1.  How to evaluate the user experience?
2.  How should the hybrid working environment be developed further?

Entry point analysis indicated the different stages with key concepts and questions. The iterations in the different stages provide a more detailed description on how the process proceed. The co-design iterations in each stage follow the five steps. During the planning-phase (P) the exiting knowledge and ideas from co-designing participants creates a plan for hybrid vision: What kind of functions will be supported by hybrid solution. It includes gathering ideas, plans, and suggestions together to create the vision. The second phase is about Artifact creation (A), where a group of co-designers creates an example based on the vision by integrating functions, technology, and place. The Evaluation phase (E) has a group of co-designers evaluate the solution. The Reflection phase (R) includes classification of intermediate feedback. The last phase is Learning (L), pointing out the outcome of the iteration cycle: What is learned in terms of digital and physical environment, including the use of both of them. It is also possible to identify further development to the possible next iteration cycle. The participants in different co-design stages are important to identify and orchestrate the combination of different stakeholders.

## 3. Results

The results are analyzed by Entry Point Analysis-tool comparing them and finally identifying the hybrid working environment.

### 3.1. Entry Point Analysis Stage 1: Hybrid Vision

### 3.1.1. Hybrid Vision Case 1

The vision of the hybrid working environment was to provide a research and collaboration platform for development of health technology in the context of medical sciences. The digital environment was an essential part of the working culture as a tool, but also as a research object. However, the vision set was not co-designed with users, it was more given by management as a solution to an operational problem of the research group: Lack of space.

The first iteration was conducted among management and included following steps:

*   Strategic meeting about development of technology research;
*   identifying the need to disperse the development work closer to its context;
*   discussing the ideas with different stakeholders;
*   identifying the locations within the city;
*   starting the negotiations with different stakeholders.

The vision was approached by focusing on location of the new solution. The iteration with users included the following steps:

*   Participatory workshop;
*   identifying the user groups and their needs;
*   reflecting on the results and starting design dialogue;

- clustering the different needs;
- identifying and mapping the requirements for digital and physical solutions.

3.1.2. Hybrid Vision of Case 2

In the second case study, the vision was a satellite campus in Namibia. It was possible to identify two iterations from this stage. The first iteration was conducted in an international group including Finnish and Namibian representatives from universities. It contained following steps:

- Participatory workshops;
- identifying the activities of the satellite campus;
- evaluating the current curricula;
- new requirements for the concept: The campus itself must be an innovative solution, not a copy from somewhere. More precisely, the ideas of novel technology supporting the presence while collaborating remotely;
- observations and new lead thoughts were listed.

This was constructing a common knowledge base. The second iteration was made in the campus of UNAM in Namibia. It contained following steps:

- Site visit;
- the identification of ICT-architecture and infrastructure requirements;
- assessment how the education can be organized;
- reflections on real infrastructure locally;
- identifying the differences of built infrastructure in the cultural context as well as better understanding the cultural differences in attitudes towards technology.

To sum up, the hybrid vision stage provided a hybrid vision in both cases. The location to realize the hybrid working environments was identified, and the collection of functional, digital and physical solutions were conducted. The vision stayed still in the abstract level. The interesting thing was that the attitudes and use of technology can play a role in the realization of future solutions.

*3.2. Entry Point Analysis Stage 2: Hybrid Integration*

3.2.1. Hybrid Integration in Case 1

This phase included iterations with users and an interior designer. Multi-functional solution and flexibility were the leading keywords in participatory workshops and in processing requirements. This was a key reason to divide the working environment for four zones and a private area. The artifact was an Activity based work environment-concept. The iteration included designing layout and interior design options, evaluating design options, and choosing the suitable version and getting the feedback ideas in design dialogue with the research group and designer. Based on classified content of feedback, the design changes were conducted.

The concept is described here more precisely, especially representing the functional zones. The hub consists of four zones, Author-zone, Lab-zone, Neighbor-zone, Synergy-zone, and two private areas that include typical technical setup. The hub has been co-designed with users and implemented for the purpose of the research group needs.

The Lab-zone provided four workstations and a possibility to test measurements with individual test persons. The Lab-zone also needed activated carbon filters and a sufficient air condition for soldering. In addition, they need flexible furniture for equipment and assembling and testing. The digital environment supported all these activities. The Author-zone was for senior researchers and group leaders with management tasks, whereas the Neighbor-zone was for younger researchers, also allowing more social interactions. The Synergy-zone mirrors the area, where researchers can meet other researchers from the building, e.g., from medical sciences. The Synergy-zone includes different kinds of conference rooms and labs that can be used as shared facilities. Figure 3 illustrates the drafts of the different zones and the final layout of the space.

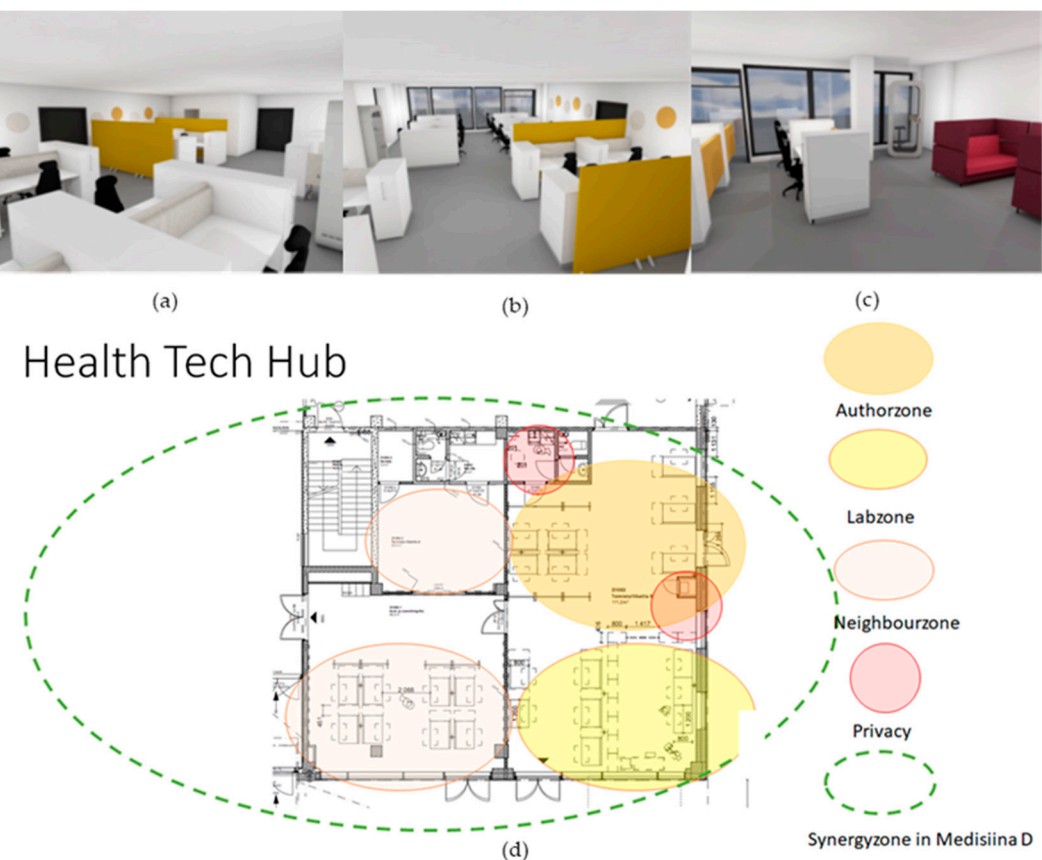

**Figure 3.** Case 1 working area divided into several zones with different activities. (**a**) Draft from the Lab-zone and Author-zone; (**b**) draft from the Author-zone to Lab-zone; (**c**) draft from the larger Neighbor-zone; (**d**) final floor plan.

3.2.2. Hybrid Integration in Case 2

Hybrid integration began in case 2 by identifying the novel technology and structural plans of the chosen space. Both technical and physical design teams initially worked separately. During the artifact creation's phase, the design team made the first drafts of the layout, dividing the space for three zones. Simultaneously, the technical team designed the novel technical setup for the place. While evaluating the drafts, multiple issues were raised. One of the issues was that technical setup would be too expensive to implement. Then some other issues were raised that needed to be taken into account. Then in the reflection phase, both physical and technical issues were specified. Good and new findings were recorded to the knowledge base in the formalization of the learning phase.

Another cycle was conducted to finalize the complete design. The technical team altered the way that technical innovation should be implemented. In the first iteration, the idea was to combine two physical spaces together with the technology by using highly expensive display walls in both university campuses. The intention was to continue the local space on the opposite space. While the technology aims to capture the physical space and everything in it, the idea changed to share any of the physical places that had the technical setup installed. So, compared to the first idea, this would be cheaper to implement and enables the possibility to share the environment for multiple locations, whereas previous was stuck to combine just two locations. In addition to this, it opened many other possibilities for learning and working. Finally, after several drafts, physical drawings were supporting the technical solution in a way that future implementation is possible after renovation.

The lab consist of three zones, the Collaborating and Co-working zone, the Welcoming zone, and the Co-learning and Connecting zone with a readiness of novel technology setup. The lab has been co-designed with stakeholders and implemented for the purpose of local

and foreign students and lectures. Figure 4 illustrates floorplan in the starting point and pictures from each zone after renovation.

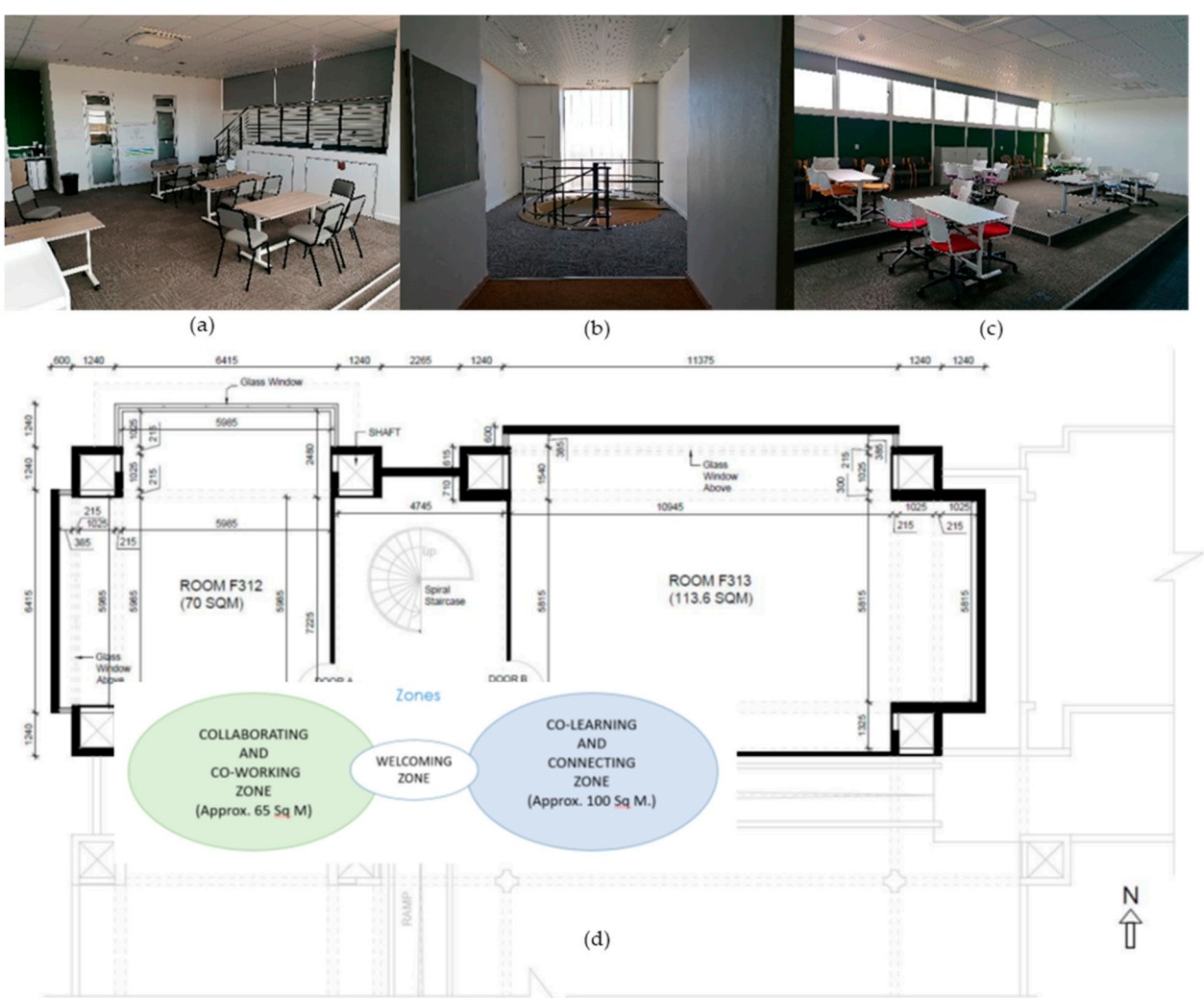

**Figure 4.** Case 2 working area divided into several zones with different activities. (**a**) Collaborating and Co-working zone; (**b**) Welcoming zone; (**c**) Co-learning and connecting zone; (**d**) floorplan in the starting point, where the unit of length is in millimeters.

The camera pair illustrated in the front of Figure 5 gives an example of the cameras used for the hybrid working environment. Fixed camera pairs will be installed on the wall around the space to be able to create a 3D constructed environment.

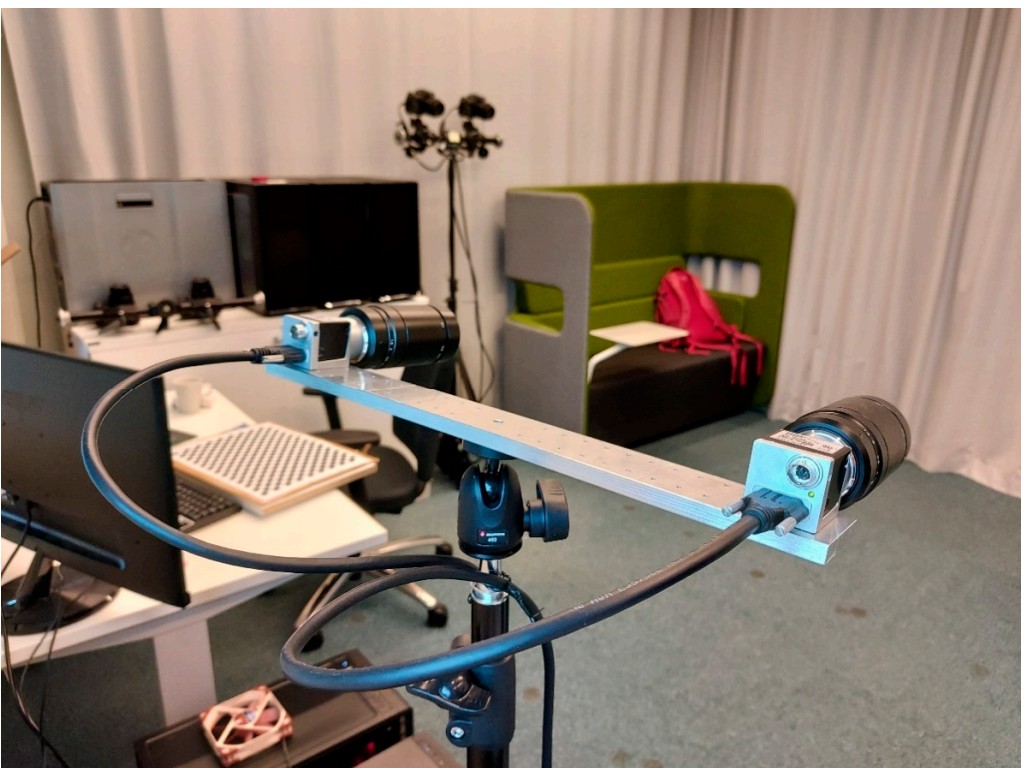

**Figure 5.** Testing setup with multiple camera pairs.

The hybrid integration stage indicated that instead of single workstations, the zoning perspective provided possibilities to discuss the digital solutions in a fluent way. The collaborative functions conducted by digital means especially bring in more requirements for physical solutions.

### 3.3. Entry Point Analysis Stage 3: Hybrid Fit

### 3.3.1. Hybrid Fit in Case 1

Hybrid fit in case 1 focused on the relocation process while the design of the layout was realized. The relocation plan and time schedule guided the user participation. However, the preliminary iteration was disrupted by the justification process, where users needed to ensure from different management levels if the relocation is really necessary. The focus was mostly on physical environment, and the potential of digital working environments was not considered. The relocation was hard to accept due the fact that the vision iterations and user need iteration were conducted with different stakeholders. However, the time schedule of removal was kept. The first impressions in settling to the new place included evaluation, and based on the instant reactions, some practical changes were made in order to improve digital and physical usability, planning scheduled with transport, furniture business, and research group, A, implementation by schedule, E, evaluating finalized workspace and getting feedback from users, R, classifying evaluation feedback and L, formalizing development suggestions.

Implementing the activity based work environment concept users to organize their work processes in the different way. The concept is based on three pillars of place (physical environment), users (behavioral environment), and technology (including knowledge sharing) where every user shares a communal workplace consisting of areas or zones that have their own purpose for certain office tasks [31]. The demand for digital collaboration increased due to the fact that the location of the physical place was different than earlier, especially in the connection with the faculty.

### 3.3.2. Hybrid Fit in Case 2

In case 2, several iterations of this stage can be identified. First, the design brief was shared with local partners in Namibia. Then, a local planning team scheduled the renovation. The artifact creation phase started the renovation, and the requirements of the novel technology were included in the project brief. The renovation was followed and reported weekly. Data, pictures, and other feedback during the process provided material for evaluation and reflection.

However, the hybrid fit stage included many small iterations, even few iterations back to design stage. One reason was the topic of power consumption: The technology needed stronger infrastructure. The 3D feed to achieve fluent digital environment required computing power. It was uncertain if the power production of the old building would be sufficient. Figure 6 illustrates the example case, how the power consumption topic was dealt.

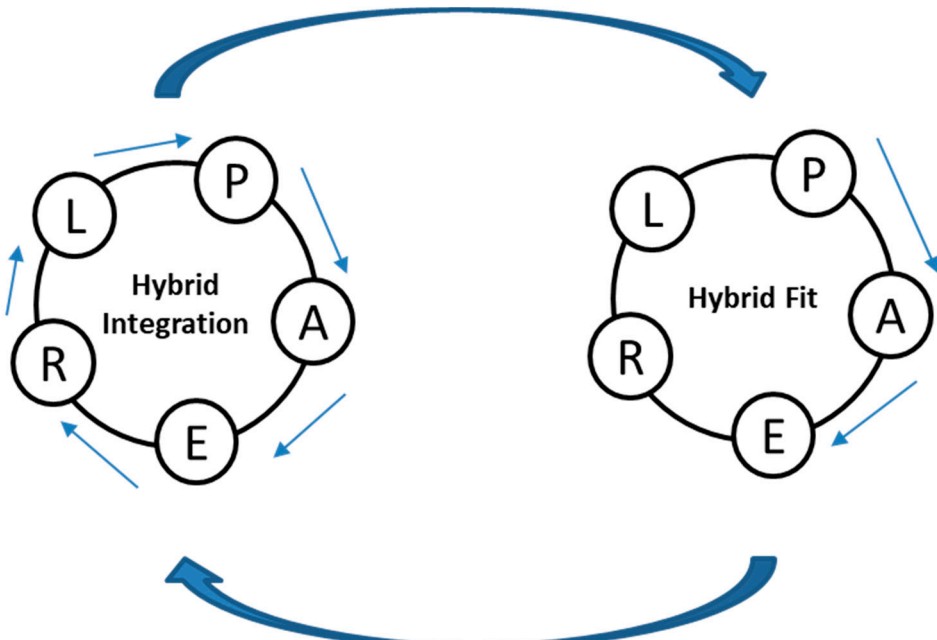

**Figure 6.** Case 2 iterations proceeded back and forth when solving the power production demanded by the digital environment.

The hybrid fit iteration was based on weekly schedules and plans. In that iteration, the possible lack in power consumption was identified. There was a need to check the layout designs, and the corrections were made accordingly—one iteration in design integration was conducted to update the structural changes to layout. The Hybrid fit iteration continued with the updated plans.

In both cases, one noticed that the iterations could happen simultaneously, either hindering or enabling the core iteration. In case 2, it was important to react on the requirement of power consumption and fix the problem before it began harming the users' daily life.

### 3.4. Entry Point Analysis Stage 4: Hybrid Fix

#### 3.4.1. Hybrid Fix in Case 1

The hybrid fix stage in case 1 has been running for 18 months now. In that time, the Neighbor-zone has been enlarged. On top of this, the interior design has been slightly improved. The hybrid fix of new environment had the following steps in the iteration:

- Planning the evaluation and identification of possible changes to the usability of hybrid working environment;
- suggesting amendments for it;

- evaluating the effect of improvements;
- classifying the benefits and possible consequences;
- formalizing needed changes for workspace.

The experience mapping was made by digital user questionnaire ($n = 8$). According to the results, the distance to the former location was a dissatisfying factor. The work practices were based on face-to-face tutoring and lecturing in the former location, while research processes were conducted in the new location. The physical dissatisfaction factors were taken care of according to the feedback, but the digital procedures in the work processes were not discussed. The expected enrichment for the previous working environment and its technology was not achieved properly.

### 3.4.2. Hybrid Fix in Case 2

In case 2, the hybrid fix stage started with the handover of physical place since the technology was not ready. The use with existing portable technology made it possible to use the current space.

Experience mapping was made by observing, interviewing, and conducting a user-questionnaire ($n = 6$). The feedback was gathered and reflected. Based on the results of the interviews and questionnaire, the atmosphere at Future Tech Lab is comfortable for co-learning and co-working. Renewed space is usable and functional for a wide variety of activities. Observation indicated that the acoustic is supporting remote presence-based collaboration as well as face to face collaboration. Now the gathered data has not yet been used to develop the environment further, because there have been some delays in production and installation of the novel technology.

The hybrid fix pointed out two clusters of experiences. Experiences of presence in the hybrid working environment are discussed both from the physical presence and the digital presence perspectives. The indoor environment can support both experiences, e.g., by good acoustics. The other cluster is experience of distance. The physical distance between different locations can be "shortened" by digital collaboration.

### 3.4.3. Summary of the Results

To sum up the results, the following recommendations for co-designing hybrid working environments can be proposed:

1. The hybrid vision of the future digital and physical solution needs to also consider the attitude and ability to use the digital solution.
2. The integration of digital environment is easier to realize in functional zones of the physical environments instead of smaller spatial units, e.g., workstations.
3. The iterations in the co-design process can happen simultaneously, and they can proceed back and forth. This may hinder or enable the improvement of the final solution. The awareness of the iterations makes it easier to manage the process.
4. The experience of hybrid working environment is not only an experience of digital platform or physical space: It is more an experience of presence and distance.

The Entry Point Analysis (EPA)-tool as a framework to analyze whether the co-design of the hybrid working environments is worth developing further. This trial indicated that hybrid vision, hybrid integration, hybrid fit, and hybrid fix provide concepts and process description which can be followed by the stakeholders of digital and physical working environments, as well by users. The Entry Point Analysis-tool made it possible to identify iterations around physical and digital objects stage by stage. It is important to visualize and follow the co-design process, not only the processes of design.

The entry point analysis indicated that the needs of the users for both physical and digital working environments need to be considered in a more integrated way than traditional usability briefing proposes. The user needs the work sets requirements for the digital solution. This technical layer sets requirements for the physical environment and elements. This was noticed, especially in the case 2. The acoustic environment needed to be designed in the way that both digital and face to face collaboration have good quality for interaction.

The entry point analysis applied from the Action Design Research approach also provides an agile perspective to the development of a more static physical working environment. This means that the usability challenges can be captured early enough to avoid usability challenges. However, if the digital and physical solutions are not integrated, one can still fail with the process. The use of a digital working environment for experiencing the physical location and distance differently would have made the user experience more satisfying. The factors for sense of presence are essential when designing a hybrid working environment.

The iteration frequency is committing the users. However, the tradition of ADR-model itself does not point out as clearly the different stakeholders as the usability briefing model does. A process which distributed the co-design process appeared in case 1. The disruption occurred due to the fact that the shared vision was too weak, and one needed to check different policies before continuing the process. In addition to the ADR process, there occurred a justification process seeking reasons to disrupt the ongoing co-design. This influenced the ADR process, especially in the design and implication stages. The process gave new ideas to the above stages that have not been mentioned previously in any workshop. The user-centered approach does not guarantee the successful outcome if the stakeholder representation is too limited.

Based on the literature review and case analysis, we state that to develop hybrid working environments, one needs to combine the ADR process model and usability process model. The entry point analysis-framework provides four key concepts for that: Hybrid vision, hybrid integration, hybrid fit, and hybrid fix.

## 4. Conclusions

This study explored the co-design challenges of hybrid working environments with an attempt to understand if there is a way to integrate two co-design approaches: The co-design of a usable built environment and the co-design of a digital environment. Both approaches focus on the process and user involvement. They differ in the length and rhythm of the process, as well as in feedback loops. The aim of the research was to explore how the conceptual framework of action design research (ADR) in Information Systems can be applied when the design object is a hybrid working environment. The exploration was made by integrating the agile digital design sprints and more static progress of built environments to the Entry Point Analysis-tool to analyze the co-design processes in two case studies of hybrid working environments.

The research question of how the action design research paradigm can be applied in understanding the usability challenges in a hybrid working environment is answered by developing and testing the Entry Point Analysis (EPA)-tool as a framework to analyze the co-design of the hybrid working environment. The hybrid vision provides a starting point for co-design to understand functions of the users conducted in digital and physical environments. The hybrid integration aims to seek both digital and physical solutions simultaneously, while hybrid fit aligns the solutions to one entity. Hybrid fix is a phase for feedback. Additionally, the EPA-tool identified iterations around physical and digital objects stage by stage, and it provided more insights into challenges of co-designing both physical and digital working environments simultaneously.

The article offers an explorative approach to understand the co-design of hybrid working environments, which are increasing. It points out the need of multidisciplinary approach to capture the design tradition of both digital and physical entities. Identification of integrated approach requires conceptual and contextual research, which is crossing the boarders of traditional design approaches.

Practical contribution is based on the result, which provided input to the co-design of hybrid working environments. It is not a question only of the hybrid solutions, but also the ability and skills to use it. The hybrid working environment is easier to understand as functional zones in the physical environment than only places which are enriched with technology. The co-design of hybrid working environments is a complex process where

one need to identify the steps taken forward and backward, as the process is not a linear path. All in all, a hybrid working environment is an experience. By co-designing the digital and physical working environment, one is co-designing the experience of presence and distance. The hybrid working environment requires learning from users and designers to identify not only the needs of the users for hybrid working environments, but also the competences to use the digital and physical solutions for different functions and purposes.

The findings of this study must be seen in the light of some limitations. The first is the number of cases. The second limitation concerns the size of cases. Both limitations effect the generalization of the results. Nonetheless, these results must be interpreted with caution of the limitations. Additionally, the more objective approach to process analysis increases the reliability. The analysis was made by researchers representing both a real estate approach and an information technology approach. More interdisciplinary backgrounds could have brought more insights to the analysis. The EPA-tool requires more validation.

The future studies focusing on the success of the hybrid working environment are needed. The impact of co-design and use of the EPA-tool as a tool to guide the process is an interesting topic for both case studies and for a longitudinal research approach. The amount of hybrid working environments are increasing, the experts of built environments and digital environments will work more closely in the future, both in practice and research. The provision of healthy and sustainable hybrid working environments requires new insights. This exploration is one step in that direction.

**Author Contributions:** Conceptualization, M.L. and S.N.; methodology, M.L. and S.N.; validation, M.L.; data analysis, M.L.; writing, M.L.; review and editing, S.N.; visualization, M.L.; supervision, S.N.; All authors have read and agreed to the published version of the manuscript.

**Funding:** This research received no external funding.

**Institutional Review Board Statement:** Not applicable.

**Informed Consent Statement:** Not applicable.

**Data Availability Statement:** Data sharing not applicable.

**Conflicts of Interest:** The authors declare no conflict of interest.

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
