# Peer review of "Design Science and Co-Designing of Hybrid Workplaces"

_buildings, doi:10.3390/buildings11030129_

Round 1
Reviewer 1 Report
The paper addresses the problem of exploration how the conceptual framework of design-science research in Information Systems can be applied assuming that the design object is a hybrid working environment.
The research topic covered by the paper can be of potential interest for the readers of this journal Buildings
The title describes the article.
The abstract reflects the content of the article, however it should be rebuild. It should be a compact part with no numbers of sections or division.
The applied methodology is introduced. However, the manuscript can be further improved to improve its informative value for further research.
It is possible to improve especially the part of the discussion and conclusion.
References should be broaden.
I recommend carefully checking the document template.
I recommend the editing of English.
Author Response
Thank you for the detailed and constructive feedback. It has been essential to improve the paper.
The following detailed changes have been made:
The abstract reflects the content of the article, however it should be rebuild. It should be a compact part with no numbers of sections or division.
Abstract has been rewritten and numbers has been removed.
The applied methodology is introduced. However, the manuscript can be further improved to improve its informative value for further research.
Major changes has been done in the method chapter
It is possible to improve especially the part of the discussion and conclusion.
Discussion and conclusion parts were rewritten and mostly the whole article
References should be broaden.
More references has been listed as suggested.
I recommend carefully checking the document template.
Document template have been checked.
I recommend the editing of English.
Minor language check done. More time for major check asked from the editor.
- The role of Design Science Research has been reduced in theoretical discussion to crystallize the compared perspectives of usability of built environment (Usability briefing approach) and digital environment (Action research design approach). The consequence of this change is that some of the tables in the former version of the article has removed. This improved the balance and structure of the paper.
- The method and analysis of the results has been rewritten. The two case studies are presented more comprehensive way. Additionally, the analysis tool, Entry point analysis tool has been presented and it provides now a comprehensive way to compare two co-design processes of hybrid environments in case studies. This changed the method and results chapter in major way. The intention has been to clarify the way how the processes have been analyzed in case studies and how the case study data is used. Based on more thorough and transparent analysis it was also possible to point out less general results. Now the results contribute more clearly to the context of hybrid environments.
The change in content and structure had the consequences to abstract and introduction chapter, allocation of the tables and figures and also conclusions and discussion.
There are some changes in the text due to this. For example Figure 2. and Table 1. has been removed from the original manuscript.
Reviewer 2 Report
The article intends to analyze the implementation of the Usability Briefing and Design Science processes in two case studies. However, it is not clear the benefit brought by these processes to the solutions presented, compared with their non-use. That is, in pictures 1 and 2 (both of which are poor quality, and which should be figures 4 and 5, respectively) it is not clear what would be different if the above-mentioned processes were not used. Still in Picture 1 (which should be figure 4) what is the Synergy-zone? Is it all space?
There are other aspects to improve significantly in the article:
- Keywords: Some of them are unsuitable, e.g. digital, hybrid, methods
- Introduction: the presentation of the structure of the article is very incomplete.
- Background: needs further development
- Case studies description: The case studies are poorly presented, e.g. in case study 2, only in the results, we understand that two different communities are involved.
- Discussion: needs further development and must include the limitations of the study.
Other formal aspects to be corrected are:
- Fig. 3: must be figure 2 and included after “Each stage in ADR process model are based on DRS model, but presented as intervention cycles including different iterations as described later.”
- Include the paragraph “Through the research we used an ADR process model. Figure 3. describes multiple iterations in ADR intervention cycles in every ADR stage. The four stages starts with Diagnosis and follows as Design, Implementation and Evolution in that order. There are five iterations, Planning ( P ), Artifact Creation ( A ), Evaluation( E ), Reflection ( R ) and Formalisation of Learning ( L ) (Mullarkey & Hevner 2019).” after this statement “Each stage in ADR process model are based on DRS model, but presented as intervention cycles including different iterations as described later.”
- Include the statment “Table 1. show each stages entry point and its’ meaning, which has been used for comparison of the case studies.” after the paragraph “Through the research we used an ADR process model. Figure 3. describes multiple iterations in ADR intervention cycles in every ADR stage. The four stages starts with Diagnosis and follows as Design, Implementation and Evolution in that order. There are five iterations, Planning ( P ), Artifact Creation ( A ), Evaluation( E ), Reflection ( R ) and Formalisation of Learning ( L ) (Mullarkey & Hevner 2019).”
- Table 1: include this Table after the paragraph “Through the research we used an ADR process model. Figure 3. describes multiple iterations in ADR intervention cycles in every ADR stage. The four stages starts with Diagnosis and follows as Design, Implementation and Evolution in that order. There are five iterations, Planning ( P ), Artifact Creation ( A ), Evaluation( E ), Reflection ( R ) and Formalisation of Learning ( L ) (Mullarkey & Hevner 2019).”
- Fig. 2: must be figure 3 and included after “Information System Research Framework can be applied in Hybrid System/working and learning environment research by including it to each ADR stage. While one can use framework to model development processes while designing physical learning environment, whereas one can use ISR (see Figure 2.) for modelling processes to co-design hybrid learning environment.”
- Please standardize references, e.g. Sonntag, Albuquerque, Magnor and Bodensiek [5] in which the 4 authors are mentioned, but when there are only 3 authors it is referred as Hilli, et al. [9].
Author Response
Thank you for the detailed and constructive feedback. It has been essential to improve the paper.
The following detailed changes have been made:
- Fig. 3: must be figure 2 and included after “Each stage in ADR process model are based on DRS model, but presented as intervention cycles including different iterations as described later.”
Figure number and location changed. Some figures have been added and some have been deleted during revision.
- Include the paragraph “Through the research we used an ADR process model. Figure 3. describes multiple iterations in ADR intervention cycles in every ADR stage. The four stages starts with Diagnosis and follows as Design, Implementation and Evolution in that order. There are five iterations, Planning ( P ), Artifact Creation ( A ), Evaluation( E ), Reflection ( R ) and Formalisation of Learning ( L ) (Mullarkey & Hevner 2019).” after this statement “Each stage in ADR process model are based on DRS model, but presented as intervention cycles including different iterations as described later.”
Paragraph have been deleted.
- Include the statement “Table 1. show each stages entry point and its’ meaning, which has been used for comparison of the case studies.” after the paragraph “Through the research we used an ADR process model. Figure 3. describes multiple iterations in ADR intervention cycles in every ADR stage. The four stages starts with Diagnosis and follows as Design, Implementation and Evolution in that order. There are five iterations, Planning ( P ), Artifact Creation ( A ), Evaluation( E ), Reflection ( R ) and Formalisation of Learning ( L ) (Mullarkey & Hevner 2019).”
Paragraph have been deleted.
- Table 1: include this Table after the paragraph “Through the research we used an ADR process model. Figure 3. describes multiple iterations in ADR intervention cycles in every ADR stage. The four stages starts with Diagnosis and follows as Design, Implementation and Evolution in that order. There are five iterations, Planning ( P ), Artifact Creation ( A ), Evaluation( E ), Reflection ( R ) and Formalisation of Learning ( L ) (Mullarkey & Hevner 2019).”
Table 1 have been deleted and entry points have been listed and described after figure 2.
- Fig. 2: must be figure 3 and included after “Information System Research Framework can be applied in Hybrid System/working and learning environment research by including it to each ADR stage. While one can use framework to model development processes while designing physical learning environment, whereas one can use ISR (see Figure 2.) for modelling processes to co-design hybrid learning environment.”
Figure numbers and location changed.
- Please standardize references, e.g. Sonntag, Albuquerque, Magnor and Bodensiek [5] in which the 4 authors are mentioned, but when there are only 3 authors it is referred as Hilli, et al. [9].
Reference style has been edited and now references are shown similar way.
- Keywords: Some of them are unsuitable, e.g. digital, hybrid, methods
Keywords has been changed.
- Introduction: the presentation of the structure of the article is very incomplete.
Structure has been now presented in the end part of introduction.
- Background: needs further development
Background has been developed further as suggested.
- Case studies description: The case studies are poorly presented, e.g. in case study 2, only in the results, we understand that two different communities are involved.
Both case description has been rewritten to clarify the situation in both case studies
- Discussion: needs further development and must include the limitations of the study.
Discussion part has also been rewritten and limitations included in the last part of the article.
Based on the excellent feedback I made larger structural changes.
- The role of Design Science Research has been reduced in theoretical discussion to crystallize the compared perspectives of usability of built environment (Usability briefing approach) and digital environment (Action research design approach). The consequence of this change is that some of the tables in the former version of the article has removed. This improved the balance and structure of the paper.
- The method and analysis of the results has been rewritten. The two case studies are presented more comprehensive way. Additionally, the analysis tool, Entry point analysis tool has been presented and it provides now a comprehensive way to compare two co-design processes of hybrid environments in case studies. This changed the method and results chapter in major way. The intention has been to clarify the way how the processes have been analyzed in case studies and how the case study data is used. Based on more thorough and transparent analysis it was also possible to point out less general results. Now the results contribute more clearly to the context of hybrid environments.
The change in content and structure had the consequences to abstract and introduction chapter, allocation of the tables and figures and also conclusions and discussion.
There are some changes in the text due to this. For example Figure 2. and Table 1. has been removed.
Reviewer 3 Report
The paper aims to explore how the conceptual framework of design science research in Information Systems can be applied when the design object is a hybrid working environment.
The objective of the article is interesting and its contents are sound.
However:
- The analysis of the state of the art is limited to describe researches focused on the topic of educational working space. I suggest adding more references to the specific case study of the hybrid working environment for office buildings.
- The methodology applied is not well explained.
- The case studies chosen are limited in size and not well presented in terms of images and description. I suggest adding more pictures of the design phase and scheme or diagrams to explain and compare better the application of ADR process entry points stage by stage.
- Discussion and Conclusion should be better presented because the present form is not sufficient to support the paper's arguments.
Author Response
Thank you for the detailed and constructive feedback. It has been essential to improve the paper.
The following detailed changes have been made:
- The analysis of the state of the art is limited to describe researches focused on the topic of educational working space. I suggest adding more references to the specific case study of the hybrid working environment for office buildings.
Hybrid working environment meaning hybrid workspace is now describe a bit more and references listed accordingly.
Analyze part has been rewritten and how the analyze have been done by using EPA tool
- The methodology applied is not well explained.
Method part has been rewritten and more well explained.
- The case studies chosen are limited in size and not well presented in terms of images and description. I suggest adding more pictures of the design phase and scheme or diagrams to explain and compare better the application of ADR process entry points stage by stage.
Case figures and description has been modified. Now figures are explained and both of the cases are described little further.
- Discussion and Conclusion should be better presented because the present form is not sufficient to support the paper's arguments.
These parts has been also rewritten and presented more clearly to support the arguments.
Based on the excellent feedback I made larger structural changes.
- The role of Design Science Research has been reduced in theoretical discussion to crystallize the compared perspectives of usability of built environment (Usability briefing approach) and digital environment (Action research design approach). The consequence of this change is that some of the tables in the former version of the article has removed. This improved the balance and structure of the paper.
- The method and analysis of the results has been rewritten. The two case studies are presented more comprehensive way. Additionally, the analysis tool, Entry point analysis tool has been presented and it provides now a comprehensive way to compare two co-design processes of hybrid environments in case studies. This changed the method and results chapter in major way. The intention has been to clarify the way how the processes have been analyzed in case studies and how the case study data is used. Based on more thorough and transparent analysis it was also possible to point out less general results. Now the results contribute more clearly to the context of hybrid environments.
The change in content and structure had the consequences to abstract and introduction chapter, allocation of the tables and figures and also conclusions and discussion.
There are some changes in the text due to this. For example the Figure 2. and Table 1. has been removed.
Reviewer 4 Report
1) No reference, especially new reference should be cited in the abstract. References [1], [2] and [3] are cited firstly in the abstract. I advise citing these only in the manuscript, not in the brief summary (abstract), hence revise the abstract. While revising, I advise skipping the numbering within the summary.
2) I advise reconsidering the keywords, terms like methods or hybrid are not help search engines to find your paper based on its content or give new and important information that can't be found in the title or abstract.
3) I advise using rather subchapters for topics that belong rather the same chapter. E.g. Introduction, background and the goal of research chapters should be parts of a single introduction chapter, while I feel that chapter 4 maybe belongs to introduction or methods, as well as an individual discussion chapter only shows that how short is that discussion on the results, so I'd prefer results and discussion chapter instead.
4) Figures called pictures... please name them figures, too. Please add proper caption which describes what we see on the figures. If a figure is a montage of different figures, please note those too with letters within the figure and add a description in the caption. E.g. in the case of Picture 1, 3 figures are on the top, it contains a floorplan too and some legend. It does not describe what are the top images show, which parts (zones) are those, where are they exactly, what is the added value to the research, etc. Same opinion apples to Picture 2, respectively.
5) Please revise the results chapter and use figures to illustrate and demonstrate these results. Besides I feel the need for illustrations, I would like to know more about the survey that is mentioned. In the methodology part of the manuscript, it is only briefly mentioned, however, I think in a scientific article, details should be given about the survey if the results and conclusions are based on that survey since the results and conclusions should be based on scientific research methods (a well constructed and evaluated survey, at least), not on opinions or told feedbacks. I don't find the research described in the paper scientific, its methodology and findings remained obscure to me.
6) Please elaborate on the new scientific contribution and findings in the manuscript if you resubmit or revise. I feel that the conclusions in the manuscript are now not more than general good advice for designers, that many of them are already following, like involve users, versatile stakeholder group, integrative and iterative design.
Author Response
Thank you for the detailed and constructive feedback. It has been essential to improve the paper.
The following detailed changes have been made:
- No reference, especially new reference should be cited in the abstract. References [1], [2] and [3] are cited firstly in the abstract. I advise citing these only in the manuscript, not in the brief summary (abstract), hence revise the abstract. While revising, I advise skipping the numbering within the summary.
References have been removed in the abstract and numbers are removed from the abstract.
I advise reconsidering the keywords, terms like methods or hybrid are not help search engines to find your paper based on its content or give new and important information that can't be found in the title or abstract.
Keywords has been changed.
- I advise using rather subchapters for topics that belong rather the same chapter. E.g. Introduction, background and the goal of research chapters should be parts of a single introduction chapter, while I feel that chapter 4 maybe belongs to introduction or methods, as well as an individual discussion chapter only shows that how short is that discussion on the results, so I'd prefer results and discussion chapter instead.
Structure of the article has been rewritten.
- Figures called pictures... please name them figures, too. Please add proper caption which describes what we see on the figures. If a figure is a montage of different figures, please note those too with letters within the figure and add a description in the caption. E.g. in the case of Picture 1, 3 figures are on the top, it contains a floorplan too and some legend. It does not describe what are the top images show, which parts (zones) are those, where are they exactly, what is the added value to the research, etc. Same opinion apples to Picture 2, respectively.
Pictures, now figures have been explained as proposed
- Please revise the results chapter and use figures to illustrate and demonstrate these results. Besides I feel the need for illustrations, I would like to know more about the survey that is mentioned. In the methodology part of the manuscript, it is only briefly mentioned, however, I think in a scientific article, details should be given about the survey if the results and conclusions are based on that survey since the results and conclusions should be based on scientific research methods (a well constructed and evaluated survey, at least), not on opinions or told feedbacks. I don't find the research described in the paper scientific, its methodology and findings remained obscure to me.
Result has been revised and illustrated more clearly. Also the structure of the later part of the article have been changed.
- Please elaborate on the new scientific contribution and findings in the manuscript if you resubmit or revise. I feel that the conclusions in the manuscript are now not more than general good advice for designers, that many of them are already following, like involve users, versatile stakeholder group, integrative and iterative design.
Contribution and findings added as suggested.
Based on the excellent feedback I made larger structural changes.
- The role of Design Science Research has been reduced in theoretical discussion to crystallize the compared perspectives of usability of built environment (Usability briefing approach) and digital environment (Action research design approach). The consequence of this change is that some of the tables in the former version of the article has removed. This improved the balance and structure of the paper.
- The method and analysis of the results has been rewritten. The two case studies are presented more comprehensive way. Additionally, the analysis tool, Entry point analysis tool has been presented and it provides now a comprehensive way to compare two co-design processes of hybrid environments in case studies. This changed the method and results chapter in major way. The intention has been to clarify the way how the processes have been analyzed in case studies and how the case study data is used. Based on more thorough and transparent analysis it was also possible to point out less general results. Now the results contribute more clearly to the context of hybrid environments.
The change in content and structure had the consequences to abstract and introduction chapter, allocation of the tables and figures and also conclusions and discussion.
There are some changes in the text due to this. For example Figure 2. and Table 1. has been removed.
Round 2
Reviewer 3 Report
After the revision, the paper show an appropriate research design. The methods are adequately described and the result clearly presented.
Author Response
Thank you for your review in both phases.
Marko
Reviewer 4 Report
Dear Authors,
1) The manuscript was hard to read due to the change tracking and due to lots of strikethroughs, deleted, reformatted part. I may advise next time only indicate the new parts with colour, and delete the deleted parts (while as supplementary material, change tracked manuscript can be also uploaded).
2) Please use all the instructions from the template of the journal (https://www.mdpi.com/files/word-templates/buildings-template.dot). Tables are still edited differently than required by the template. However, figures now contain the necessary captions, but I advise that the captions within a figure should not hang over each other
3) Conclusion chapter should be focused and include a summary of the actual results of the research, not just generalizations or the presentation of that authors used a different/new methodology. I miss the novelty contribution and the originality of the paper in the present form.
4) The questionnaires, especially the questions could be published in an appendix. Now the paper includes questions asked in a different, so-called hybrid vision stage. It did not turn out to me that these questions were the same as the above mentioned 26 or 32 question containing questionnaires.
5) Although the number of responses was very low, and the paper does not give any perspective on what kind of people participated in the workshops or who responded to the questionnaires. So basically the paper describes a summary from the authors perspective on the topic according to or based on 8 and 6 peoples' opinion. It is not scientific at all.
Author Response
Dear reviewer,
I want to thank you again for your feedback. Bellow, each step have been explained and changed text has green colour in the manuscript.
1) The manuscript was hard to read due to the change tracking and due to lots of strikethroughs, deleted, reformatted part. I may advise next time only indicate the new parts with colour, and delete the deleted parts (while as supplementary material, change tracked manuscript can be also uploaded).
The submitted version includes the changes to the second version of the paper to make it more readable
2) Please use all the instructions from the template of the journal (https://www.mdpi.com/files/word-templates/buildings-template.dot). Tables are still edited differently than required by the template. However, figures now contain the necessary captions, but I advise that the captions within a figure should not hang over each other
Tables have been fixed and slightly modified (who participated in workshops and questionnaires) and figures have changed without hang effect. Article should now follow the templates rules.
3) Conclusion chapter should be focused and include a summary of the actual results of the research, not just generalizations or the presentation of that authors used a different/new methodology. I miss the novelty contribution and the originality of the paper in the present form.
Conclusion chapter is rewritten to emphases more the actual results.
4) The questionnaires, especially the questions could be published in an appendix. Now the paper includes questions asked in a different, so-called hybrid vision stage. It did not turn out to me that these questions were the same as the above mentioned 26 or 32 question containing questionnaires.
The themes of the questionnaires were similar as mentioned in the text. The adding of the structure of the questionnaire is now also in the text. Additionally, the research design is now explained more in details emphasising more the qualitative approach,
“ Data gathering focused on two topics: the co-design process of hybrid working environment and the user experience of the hybrid working environment. The aim of the study was to how to integrate the design of digital and physical to achieve the hybrid working environment, which is usable. The qualitative approach to the problem seems most relevant because the co-design of either physical or digital environment is more typical than the integrated approach. The quantitative data is not easy to gather because the hybrid working environments are not yet the usual case in the context of working.
The data collection was performed in three steps.
First step was based on literature review. It was a basis to the integrated Entry point analysis (EPA) tool. The tool aims to identify the different phases of co-designing the hybrid working environments
The second step was dedicated to the analysis of the co-design processes. The data was gathered by participatory workshops and document analysis. ….
In the third step of data gathering focused on user experiences. Data was gathered by observations and experience mapping questionnaires.”
5) Although the number of responses was very low, and the paper does not give any perspective on what kind of people participated in the workshops or who responded to the questionnaires. So basically the paper describes a summary from the authors perspective on the topic according to or based on 8 and 6 peoples' opinion. It is not scientific at all.
Added explanation, who participated in workshops and questionnaires.
Triangulation was added to the text
The intention of the questionnaires was not to produce generalized quantitative data, but serve as one sources of data next to the observations, participatory workshops and document analysis. Triangulation (Knali and Breitmayer ) was used to complement the data and to find new information, in other words to get additional pieces to the overall picture of hybrid working environment.
Round 3
Reviewer 4 Report
Dear Authors,
I think the manuscript is improved significantly due to the revisions. I may advise improving the keywords before publishing the paper.
I recommend adding "Action Design Research" or ADR, and "Entry Point Analysis" or EPA, and maybe "project management" into the keywords. Instead of "workplace", I recommend "hybrid working environment" as a keyword, since "workplace" is in the title as well. I may extend "usability" to "usability briefing" as a keyword. Therefore I accept the paper to be published after these minor, optional revisions.
Author Response
Dear reviewer,
Suggested changes for the keywords has been made. Thank you for putting so much effort on this article and all the feedback.
Kind regards,
Marko